# Analysis of Asphaltene Precipitation Models from Solubility and Thermodynamic-Colloidal Theories

**Esaú A. Hernández** 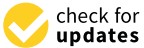**, Carlos Lira-Galeana and Jorge Ancheyta ***

Instituto Mexicano del Petróleo, Eje Central Lázaro Cárdenas Norte 152, Col. San Bartolo Atepehuacan, Mexico City 07730, Mexico
* Correspondence: jancheyt@imp.mx

**Abstract:** Asphaltenes are known to cause problems related to flocculation, precipitation, and plugging, either in the formation, production lines, and processing equipment. Different models have been proposed to predict the thermodynamic conditions under which asphaltenes precipitate over the past years. This work analyses the performance of various models on their capability to match the literature experimental data of precipitated asphaltene mass fractions. Twenty-five different models based on equation-of-state (EoS), polymer solution, and thermodynamic-colloidal theories were identified. The performance/test datasets were collected and classified according to their pressure/temperature conditions, $CO_2$, n-$C_5$/n-$C_7$ gas, and liquid titrations. Statistical analysis, including residuals, parity plots, and average absolute relative deviation (AARD, %), were used to compare the adequacy of selected models. Results confirmed the need for further model development for general applications over wide pressure, temperature, and composition intervals.

**Keywords:** asphaltene precipitation modeling; EoS; polymer solutions; colloidal theories; model evaluation

## 1. Introduction

The growing production of heavy crude oil as reservoirs continue to deplete has motivated the search for methods to improve their extraction, transportation, and refinement. One of the natural properties of these oils is their high amount of asphaltenes, which causes high viscosity and difficulties in producing and refining them.

Asphaltenes are a solubility class of compounds with high aromaticity, high molecular weight, and an undefined boiling point. Asphaltenes are soluble in aromatic compounds, such benzene or toluene, and insoluble in low-molecular-weight alkanes, such as n-pentane or n-heptane. Asphaltenes are considered the most polarizable and aromatic fraction of crude oil. They are rich in heteroatoms (nitrogen, oxygen, and sulfur) and metals (nickel and vanadium) [1]. Asphaltenes mainly exist as monomers in bulk crude oil, while they behave as a polymer upon association and precipitation [2]. In both refinery and production operations, asphaltene dispersion with chemicals is preferred to avoid their aggregation and subsequent precipitation. It is generally accepted that resins, which are absorbed on the surface of asphaltenes, are natural peptizing agents of asphaltenes [3]. Asphaltenes are known to be the main precursor of sediment formation, with major difficulties in oil production, transportation, and processing equipment [4]. Several investigations have revealed that asphaltene behavior is influenced by pressure, temperature, crude oil properties, type and amount of precipitant, and characteristics of porous media (oil wells). Solid formation inducted by asphaltene precipitation causes major effects on production systems, in both upstream and downstream operations. [5].

Asphaltene precipitation is influenced by the nature of the medium in which they are hosted [6]. Changes in composition (in-field mixing with different crude oils, addition of solvent, dispersant, $CO_2$ injection, etc.) can modify the stable-to-unstable conditions in the

medium, leading heavy organics to flocculate, precipitate, and deposit [4]. Asphaltene adsorption onto surfaces is a phenomenon that can be used to predict and avoid asphaltene precipitation from reservoir production and downstream operations. Asphaltene adsorption could be an efficient method to assist enhanced oil recovery efforts [7].

Rheological properties of the oil are important for the study of precipitation of asphaltenes in different stages of the oil production chain. Asphaltene flow behavior in diverse media, including waxy matrix, polymer matrix, and oil/water emulsions, represents interactions at the interface between asphaltene-asphaltene, asphaltene-maltenes, and asphaltene-water. Because the amount of asphaltene and resins tends to increase as a result of oil well decline, rheological properties of oil changes and asphaltene behavior is more complex and tends to precipitate onto reservoir and production facilities [8].

Therefore, the study of asphaltenes is of great importance for anticipating the problems that they may cause, particularly when dealing with unstable, asphaltenic crude oils. In previous works, our group reviewed methods based on SARA (saturates, aromatics, resins asphaltenes) analysis to determine the stability of crude oils [9]. We applied them to a wide range of Mexican crude oils [10]. Apart from SARA-analysis-based methods, there are other more sophisticated models which use a solubility approach or a colloidal approach to predict precipitation.

In this work, we perform a comprehensive review of the literature models used to calculate asphaltene precipitation, either as a function of pressure and temperature, or from gas and liquid titration data, using their reported data-matching accuracy and statistical tests. Conclusions on the most appropriate models are given, based on the above tests.

## 2. Description of Models for Asphaltene Precipitation

To estimate asphaltene precipitation, various studies have been carried out using different approaches, and these can be classified into two different approaches: solubility approach models and colloidal approaches models [11].

### 2.1. Solubility Approach Models

It is common to quantitatively describe asphaltene precipitation via parameters of solubility. The solubility parameter indicates the relative solvency behavior of a specific solvent, and the relationship among solubility, van der Waals forces, and the cohesive energy density. Hildebrand defined the solubility parameter as the square root of the cohesive energy density (Vargas and Tavakkoli 2018).

The solubility approach is classified into four different theories: regular solution theory models (RST), cubic equation of state models (C-EoS), cubic plus association equation of state models (CPA-EoS), statistical association fluid theory equation of state models (SAFT-EoS), Scott-Magat theory models (SMT), and Flory-Huggins theory models (FHT). Table 1 summarizes the RST models, while C-EoS, CPA-EoS, SAFT-EoS, SMT, and FHT are reported in Tables 2 and 3.

**Table 1.** Models to calculate asphaltene precipitation based on solubility approach (RST).

| Regular Solution Theory Author | Equation | Author | Equation |
|---|---|---|---|
| (1) Hirschberg et al. | $(x_{va})_{max} = exp\left\{ \frac{V_a}{V_L}\left[1 - \frac{V_L}{V_a} - \frac{V_L}{RT}(\delta_a - \delta_L)^2\right]\right\}$ | (6) Chung | $x_a^L = exp\left[-\frac{\Delta h_f^a}{RT}\left(1 - \frac{T}{T_f^a}\right) - \frac{V_a^L}{RT}(\delta_m^L - \delta_a^L)^2 - \ln\frac{V_a^L}{V_m} - 1 + \frac{V_a^L}{V_m}\right]$ |
| (2) Burke et al. | $x_{va} = exp\left\{ \frac{V_a}{V_L}\left[1 - \frac{V_L}{V_a} - \frac{V_L}{RT}(\delta_a - \delta_L)^2\right]\right\}$ | (7) Cimino et al. | $\ln(1 - x_{va}^*) + \left(1 - \frac{V_S}{V_a}\right)x_{va}^* + \frac{V_S}{RT}(\delta_a - \delta_L)^2 x_{va}^{*2} = 0$ |
| (3) Novosad and Constain | $v_i^{(k)} = exp\left[1 - \frac{V_i}{V^{(k)}} - \frac{V_i}{RT}\left(\delta_i - \delta^{(k)}\right)^2\right]$ | (8) de Boer et al. | $S = exp\left\{-1 + V_a\left[\frac{1}{V_o} - \frac{(\delta_a - \delta_o)^2}{RT}\right]\right\}C$ |
| (4) Rassamdana et al. | $x_{va}^L = x_{va}^S exp\left[\frac{V_a^L}{V^L} - 1 - \frac{V_a^L}{RT}(\delta_a - \delta_L)^2\right]$ | (9) Alboudwarej et al. | $K_i = \gamma_i^{L_1} = exp\left\{\ln\frac{V_i^{L_1}}{V_m^{L_1}} + 1 - \frac{V_i^{L_1}}{V_m^{L_1}} + \frac{V_i^{L_1}}{RT}(\delta_i - \delta_m)^2\right\}$ |
| (5) Buckley et al. | $\delta = \left(\frac{\sqrt{3}\pi}{384}\frac{hV_a}{\sigma^3}\right)^{1/2}\frac{\sigma^3}{V/N_A}\frac{n^2-1}{(n^2+2)^{3/4}}$ | (10) Yarranton and Masliyah | $K_i = exp\left\{\frac{\Delta H_i^f}{RT}\left(1 - \frac{T}{T_i^f}\right) + 1 - \frac{V_i^L}{V_m} + \ln\left(\frac{V_i^L}{V_m}\right) + \frac{V_i^L}{RT}(\delta_m - \delta_i^L)^2\right\}$ |
| (11) Thomas et al. | $K_i^S = \frac{x_i^S}{x_i^L} = exp\left\{\frac{V_i}{RT}\left[(\delta - \delta_i)_L^2 - (\delta - \delta_i)_S^2\right] + \frac{\Delta H_f}{RT}\left[1 - \frac{T}{T_f}\right] + \frac{\Delta C_P}{R}\left[1 - \frac{T_f}{T} + \ln\frac{T_f}{T}\right] + \int_0^P\left(\frac{V_i^L - V_i^S}{RT}\right)dP\right\}$ | | |
| (12) Wang and Buckley | $\ln(1 - x_{va}^L) + \left(1 - \frac{V_m}{V_a}\right)x_{va}^L + \mathcal{X}(x_{va}^L)^2 = \ln(1 - x_{va}^H) + \left(1 - \frac{V_m}{V_a}\right)x_{va}^H + \mathcal{X}(x_{va}^H)^2$ <br> $\ln x_{va}^L + (1 - x_{va}^L)\left(1 - \frac{V_a}{V_m}\right) + (1 - x_{va}^L)^2\frac{V_a}{V_m}\mathcal{X} = \ln x_{va}^H + (1 - x_{va}^H)\left(1 - \frac{V_a}{V_m}\right) + (1 - x_{va}^H)^2\frac{V_a}{V_m}\mathcal{X}$ <br> $\mathcal{X} = \frac{V_m}{RT}(\delta_a - \delta_m)^2$ | | |

**Table 2.** Models to calculate asphaltene precipitation based on solubility approach (EoS, SAFT, SAFT-HS, and SAFT-VR).

| **Equation of State (EoS)** | | | |
|---|---|---|---|
| **Author** | **Equation** | **Author** | **Equation** |
| (13) Nghiem et al. | $\ln f_a = \ln f_a^* + \frac{V_a(P-P^*)}{RT}$ | (14) Sabbagh et al. | $C_i = 0.3796 + 1.485\omega_i - 0.1644\omega_i^2 + 0.01667\omega_i^3$ |

| **Statistical Association Fluid Theory (SAFT)** | | **Statistical Association Fluid Theory-Hard Sphere (SAFT-HS)** | |
|---|---|---|---|
| **Author** | **Equation** | **Author** | **Equation** |
| | | | $A = A^{id} + A^{hs} + A^{vdw} + A^{assoc} + A^{chain}$ |
| (15) Ting et al. | $\frac{A^{res}}{RT} = \frac{A^{seg}}{RT} + \frac{A^{chain}}{RT} = m\left(\frac{A_0^{hs}}{RT} + \frac{A_0^{disp}}{RT}\right) + \frac{A^{chain}}{RT}$ <br><br> $\frac{A_0^{hs}}{RT} = \frac{6}{\pi\rho}\left[\frac{\zeta_2^3 + 3\zeta_1\zeta_2\zeta_3 - 3\zeta_1\zeta_2\zeta_3^2}{\zeta_3(1-\zeta_3)^2} - \left(\zeta_0 - \frac{\zeta_2^3}{\zeta_3^2}\right)\ln(1-\zeta_3)\right]$ <br><br> $\frac{A_0^{disp}}{RT} = \frac{A_1}{RT} + \frac{A_2}{RT}$ <br><br> $\frac{A^{chain}}{RT} = \sum x_i(1-m_i)\ln g_{ii}^{hs}(d_{ii})$ | (16) Wu et al. | $\frac{A^{id}}{kT} = \sum_{i=1}^{2} N \ln\left(\rho_i \Lambda_i^3\right) - N_t$ <br><br> $\frac{A^{hs}}{kT} = N_t\left\{\left[\frac{\xi_2^3}{\xi_0\xi_3^2} - 1\right]\ln(1-\xi_3) + \frac{3\xi_1\xi_2}{\xi_0(1-\xi_3)} + \frac{\xi_2^3}{\xi_3\xi_0(1-\xi_3)^2}\right\}$ <br><br> $\frac{A^{vdw}}{kT} = \frac{V}{2}\sum_{i=1}^{2}\sum_{j=1}^{2}\rho_i\rho_j\frac{U_{ij}}{kT}$ <br><br> $\frac{A^{assoc}}{kT} = 2N_A\left(\ln x_\alpha + \frac{1-x_\alpha}{2}\right) + N_R\left(\ln x_\beta + \frac{1-x_\beta}{2}\right)$ <br><br> $\frac{A^{chain}}{kT} = N_R(1-l_R)\ln g_{22}^{hs}(\sigma_{22})$ |

| **Statistical Association Fluid Theory-Variable Range (SAFT-VR)** | | | |
|---|---|---|---|
| **Author** | **Equation** | | |
| (17) Buenrostro et al. | $\frac{A}{NkT} = \frac{A^{ideal}}{NkT} + \frac{A^{mono}}{NkT} + \frac{A^{chain}}{NkT} + \frac{A^{assoc}}{NkT}$ <br><br> $\frac{A^{ideal}}{NkT} = \left(\sum_{i=1}^{n} x_i \ln \rho_i \Lambda_i^3\right) - 1$ <br><br> $\frac{A^{mono}}{NkT} = \left(\sum_{i=1}^{n} x_i m_i\right)\frac{A^M}{N_s kT} = \left(\sum_{i=1}^{n} x_i m_i\right)a^M$ | $\frac{A^{chain}}{NkT} = -\sum_{i=1}^{n} x_i(m_i - 1)\ln y_{ii}^M(\sigma_{ii})$ <br><br> $\frac{A^{assoc}}{NkT} = \sum_{i=1}^{n} x_i\left[\sum_{a=1}^{S_i}\left(\ln X_{\alpha,i} - \frac{X_{\alpha,i}}{2}\right) + \frac{S_i}{2}\right]$ | |

**Table 3.** Models to calculate asphaltene precipitation based on solubility approach (CPA-EoS, SMT, and FHT).

| Cubic Plus Association Equation of State | | | |
|---|---|---|---|
| **Author** | **Equation** | **Author** | **Equation** |
| (18) Li and Firoozabadi | $\frac{A^{ex}_{ph}}{nRT} = -\ln(1-b\rho) - \frac{a}{2\sqrt{2}bRT}\ln\left(\frac{1+(1+\sqrt{2})b\rho}{1+(1-\sqrt{2}b\rho)}\right)$ <br> $\frac{A^{ex}_{assoc.}}{nRT} = N_a \mathcal{X}_a\left(\ln \mathcal{X}_a + \frac{1-\mathcal{X}_a}{2}\right) + N_R \mathcal{X}_R\left(\ln \mathcal{X}_R + \frac{1-\mathcal{X}_R}{2}\right)$ | (19) Shirani et al. | $Z = Z^{ph} + Z^{assoc.}$ <br> $Z^{assoc.} = -\frac{1}{2}\left(1 + \frac{1}{V}\frac{\partial \ln g}{\partial\left(\frac{1}{V}\right)}\right)\sum_i x_i \sum_{A_i}(1-x_{a_i})$ |
| **Scott-Magat theory** | | | |
| **Author** | **Equation** | | |
| (20) Kawanaka et al. | $V^L_{f_a} = \int dV^L_{f_{a_i}} = \int_0^\infty \left\{\frac{\left(\frac{M_{a_i}}{\overline{M_a}}\right)V^c_a}{V^L + V^S exp\left(-N_{sa_i}\theta\right)}\right\}F(M_{a_i})dM_{a_i}$ | | |
| **Flory-Huggins theory** | | | |
| **Author** | **Equation** | | |
| (21) Flory-Huggins | $\frac{\Delta G_m}{RT} = x_a\ln x_{va} + x_b\ln x_{vb} + x_a x_{vb}\mathcal{X}_{ab}$ <br> $\mathcal{X}_{ab} = \frac{V_r}{RT}(\delta_a - \delta_b)^2$ | | |
| (22) Flory-Huggins-Zuo equation of state (FHZ EoS) | $\Delta A(h) = \Delta A_{entropy}(h) + \Delta A_{sol}(h) + \Delta A_{grav}(h)$ <br> $\Delta A_{entropy}(h) = kT\sum_i n_i\ln \varphi_i$ | | |

2.1.1. Regular Solution Theory

The regular solution theory was originally developed by Scatchard, Hildebrand, and Wood to describe the thermodynamics of solutions [12]. Models developed by this theory provide semiquantitative estimates of solubility parameters for solutions of nonpolar liquids. The models based on RST identified in this work are the following:

(1) Hirschberg et al., model [13]. This model was developed to describe the behavior of asphalt and asphaltenes in crude oil reservoir upon changes in pressure, temperature, or composition. The model appears to be well applicable to conditions at which asphaltenes are associated with resins, and may be used to identify field conditions where asphalt or asphaltene precipitation would occur. The model overestimates the solubility of asphaltenes at very high dilution ratios [11].

(2) Burke et al., model [14]. The model describes the precipitation mechanism as a polymer solution theory. The overall model depends on two types of fluid equilibria: V/L equilibrium of the total fluid and L/L equilibrium between liquid oil and pseudo-liquid asphaltene phases. The agglomeration of asphaltenes may hinder the quantitative performance of the model. Data generated by the model can be used to determine critical properties of the solvent/oil system. The model can also be used to estimate the probability of precipitates formation as the composition and properties of the reservoir fluid change.

(3) Novosad and Costain model [15]. Hirschberg's model with asphaltene-asphaltene and asphaltene-resin association was used to correlate asphaltene precipitation data. The Peng-Robinson EoS was used to determine the V/L equilibrium data on oil-$CO_2$ mixtures. The model has a large number of fitting parameters. More data on physical properties of asphaltenes and resins are needed to predict asphaltene stability and the extent of their precipitation with confidence. All asphaltene precipitation data were successfully correlated using a molecular thermodynamic model with association. Model calculations indicated that asphaltene destabilization may be minimized by producing wells at high wellhead flowing pressures.

(4) Rassamdana et al., model [16]. The model employs a scaling function, somewhat like those encountered in aggregation and gelation phenomena. The scaling function has a very simple form, and its predictions agree well with the experimental data. This scaling equation provides a particularly simple, and apparently universal, prediction for the onset of asphaltene (or asphalt) precipitation.

(5) Buckley et al., model [17]. The model assumes that the dominant intermolecular interaction energy governing asphaltene precipitation is the London dispersion contribution to the van der Waals forces. The interaction energy is a function of the differences between the squares of the refractive indices of the asphaltene and solvent. Solubility parameters of the asphaltene and solvent are related to their refractive indices. The refractive index is a function of the composition and density. Refractive indices were extrapolated to zero frequency as a parameter into the model.

(6) Chung model [18]. The model is based on thermodynamic principles for solid-liquid phase equilibrium and assumes that asphaltenes are dissolved in oil in a true liquid-solid state, not in a colloidal suspension. The model considers the effects of temperature, composition, and activity coefficient on the solubility of wax and asphaltenes in organic solutions, and can predict the solubility of asphaltene in crude oil systems.

(7) Cimino et al., model [19]. The model based on polymer solution thermodynamics and was developed using experimental phase behavior data. The model allows for the prediction of asphaltene stability with few experimental data, and considers that on phase separation, asphaltenes contain a fraction of the solvent.

(8) de Boer et al., model [20]. The model is based on the solubility of the oil and the asphaltenes and their molar volumes (similar to the Hirschberg model). The author found that all crude oil properties were correlated with the density of the crude at in situ conditions. The model assumes that the asphaltene precipitation depends on the degree of saturation of the asphaltene phase due to the pressure drop during production.

(9) Alboudwarej et al., model [21]. The model is based on the liquid-liquid equilibrium regular solution theory. The input parameters of the model are the mole fraction, molar volume, and solubility parameters of each component. During the model development, asphaltenes were divided into fractions with different associated molar mass according to the Schultz-Zimm molar mass distribution. The effect of the solvent type and the onset and amount of asphaltene precipitation can be calculated with the model.

(10) Yarranton and Masliyah model [22]. The model describes the asphaltene solubility using a solid-liquid equilibrium through the calculation of K-values derived from Scatchard-Hildebrand solubility theory and Flory-Huggins entropy of mixing. Asphaltenes were considered as a series of polyaromatic hydrocarbons with randomly distributed associated functional groups. The model calculates asphaltene precipitation onset and the amount of precipitated asphaltenes.

(11) Thomas et al., model [23]. The model relates the fugacities of the liquid/solid components. The model's main contribution is the correlation of the required properties: enthalpy change of fusion, fusion temperature, solubility parameters, and liquid partial molar volumes.

(12) Wang and Buckley model [24,25]. This asphaltene solubility model (ASM) was developed to predict the phase behavior of asphaltenes in crude oil. The thermodynamic model was derived from Flory-Huggins polymer theory and reproduces a wide range of experimental data for the onset of asphaltene precipitation. The better prediction of the model over others arises by the estimation of solubility parameters based on refractive indices measurements, the solution of the thermodynamic equations to obtain compositions of both asphaltene poor and asphaltene rich phases, and the use of the Gibbs free energy curve to define onset conditions.

### 2.1.2. Cubic Equation-of-State (C-EoS)

Equations of state are useful to describe properties of fluids, mixtures, and solids. In the oil industry, the most widely used EoS are Soave-Redlich-Kwong (SRK) and Peng-Robinson (PR). Equations of state are not limited to describing the liquid-vapor equilibrium, but they can also describe liquid-liquid and liquid-solid equilibria. Models developed from equations of state include the following.

(13) Nghiem et al., model [26]. The model is based on the division of the heaviest component in the oil into a non-precipitating and a precipitating component. Model can make quantitative calculations of experimental data from the literature, as well as additional data from industry. Asphaltenes are considered a pure dense phase, and are referred to as the asphalt phase and can either be liquid or solid. The model can calculate a decrease in asphaltene precipitation at high solvent concentrations.

(14) Sabbagh et al., model [27]. The model is an adaptation of the Peng-Robinson equation of state using group contribution methods for the fitting and prediction of the onset and amount of asphaltene precipitation from both asphaltenes/toluene/n-alkane and bitumen/n-alkane systems. A liquid-liquid equilibrium is assumed with only asphaltenes partitioning to the dense phase, while saturates, aromatics, and resins are considered as single pseudo-components. The model matches asphaltene yields for n-alkane diluted bitumen. However, it fails to fit yields from n-pentane-diluted bitumen at high dilution ratios.

### 2.1.3. Statistical Association Fluid Theory Equation of State (SAFT-EoS)

SAFT-EoS is the most widely used for the prediction of asphaltene phase behavior by applying Wertheim's theory. SAFT is an equation of state where the molecules are modeled in the form of chains composed of bonded spherical segments. This equation of state describes the residual Helmholtz free energy ($A^{res}$) of a mixture of associating fluids. The PC-SAFT equation of state is organized into different types of intermolecular interactions, such as the hard chain reference, dispersion, association, polar interaction, and ions.

(15) Ting et al., model [28]. The SAFT equation of state was used to model asphaltene phase behavior in live oil (mixture of n-C7 insoluble asphaltenes, toluene, and methane)

and a recombined oil (stock tank oil with its separator gas). The refractive index of the mixture at onset of asphaltene precipitation was used to characterize SAFT parameters of the asphaltenes. With this data, the densities for stock tank oil and the recombined oil were predicted very well with experimental measurements.

(16) Wu et al., model [29,30]. The author considered that asphaltenes are represented by attractive hard spheres that can associate with themselves. The Helmholtz energy of hard spheres (A^hs) was included in the SAFT equation (SAFT-HS). The author concludes that the effect of the oil medium on asphaltene precipitation is only determined by its Hamaker constant (obtained from oil´s density and concentration of light compounds in the oil).

(17) Buenrostro et al., model [31]. The author includes intermolecular interactions in SAFT EoS with variable-ranged potentials (SAFT-VR). SAFT-VR approach exploits that molecular parameter to model real effects in fluids. SAFT-VR approach resulted in a promising ability to predict phase equilibria of asphaltene precipitation due to changing conditions (P, T, and composition), as well as for pressure depletion at reservoir conditions in live oil samples.

### 2.1.4. Cubic Plus Association Equation of State (CPA-EoS)

The cubic plus association equation of state is a combination of the classical cubic equation of state and chemical contribution (association). Classical EoS describes the physical part of attraction and repulsion, and the chemical contribution is related to Chapman's association term originally developed for statistical association fluid theory (SAFT). The CPA equation of state has been successfully applied to a variety of complex phase equilibria, including mixtures containing alcohols, glycols, organic acids, water, and hydrocarbons. The following models are based on this approach:

(18) Li and Firoozabadi model [32]. It is applied to model the effects of temperature, pressure, and composition on asphaltene precipitation in live crude oils. A liquid-liquid equilibrium between the upper onset and bubble point pressures, and a gas-liquid-liquid equilibrium between the bubble point and lower onset pressures were considered to develop the model. The model's advantage is based on the existing fluid characterization, which can be readily implemented in compositional reservoir simulators. It was able to reproduce the amount and onset pressures of asphaltene precipitation in several live oils over a broad range of composition, temperature, and pressure conditions.

(19) Shirani et al., model [33]. The model is based on a combination of a physical part and an association term. The model combines a cubic EoS and association (chemical) terms from Wertheim theory. The interactions between molecules are considered in the physical and association parts. The model is expressed in terms of the compressibility factor, Z. The physical contribution of the compressibility factor was obtained using Peng-Robinson and Soave-Redlich-Kwong equations of state. For the physical part, SRK EoS gives more accurate results than the PR EoS for predicting the asphaltene phase behavior. The model showed good accuracy with experimental data from three live oil samples.

### 2.1.5. Scott-Magat Theory

The Scott-Magat theory assumes that polymers have a heterogeneous structure, and polydispersity plays an important role in the molecular weight of polymers. This assumption is applicable when using the following model.

(20) Kawanaka et al., model [34]. This statistical thermodynamic model is used to predict the onset point and amount of asphaltene deposition of crude oils. Asphaltene is assumed to consist of many components of similar polymeric molecules. The model is used to predict the phase behavior in $CO_2$/oil mixtures, and is applicable to estimate organic deposition (asphaltene, wax, diamantine, etc.) from reservoir fluids under the influence of a miscible solvent at various temperatures, pressures, and compositions.

### 2.1.6. Flory-Huggins Theory

The Flory-Huggins theory was originally developed by Flory and Huggins to describe the Gibbs free energy mixing of polymer solutions [35,36]. The theory assumes that asphaltenes have a homogeneous structure and properties. Models based on this theory are the following.

(21) Flory-Huggins model [35,36]. It assumes that the polymer has the form of a flexible chain of segments and that each segment is equal in size to a solvent molecule. Several models are based on this theory.

(22) Flory-Huggins-Zuo equation of state (FHZ EoS) [37]. Flory-Huggins-Zuo EoS assumed that a reservoir fluid is treated as a mixture with two pseudo-components: non-asphaltene (or maltene) and asphaltene components. It also describes the equilibrium concentration distribution of heavy ends in the oil column. The FHZ EoS includes gravitational forces on the existing Flory-Huggins regular solution model, which has been used to model asphaltene precipitation in the oil and gas industry. It has been successfully employed to estimate asphaltene concentrations in different crude oil columns around the world, incorporating the size of asphaltene molecules, asphaltene nanoaggregates, and asphaltene clusters. Downhole fluid analysis (DFA) has been used to measure continuous fluid profiles and properties of discontinuous fluids in reservoir connectivity [38]. DFA measurements are related to all parameters of the FHZ EoS, like composition, the gas-oil ratio, and the density of heavy components.

### 2.2. Colloidal Approach Models

The colloidal approach assumes that asphaltenes exist in the oil medium as solid particles suspended and stabilized by resins. The short-range intermolecular repulsions between resin molecules adsorbed on neighboring asphaltene particles and the long-range repulsions between asphaltene particles are the conditions for keeping asphaltenes stable in solution. On the other hand, the precipitation of asphaltenes is assumed to be an irreversible process, and a certain quantity of resins is necessary to completely peptize the asphaltenes in crude oil. The colloidal approach has three different paths to describe asphaltenes precipitation: chemical potential, micellization, and reverse micellization. Table 4 summarizes the models based on the colloidal approach.

**Table 4.** Models to calculate asphaltene precipitation based on the colloidal approach (Chemical Potential, Micellization, and Reverse Micellization).

| Chemical Potential Author | Equation | Micellization Author | Equation |
|---|---|---|---|
| (23) Leontaritis and Mansoori | $\mu^A_{resin} = \mu^o_{resin}$ <br> $\frac{\Delta \mu_R}{RT} = \frac{\mu_R - (\mu_R)_{reference}}{RT} = \ln(x_{vR}) + 1 - \frac{v_R}{v_m} + X_R$ | (24) Victorov and Firozoobadi | $x_M = x^{n1}_{a1} x^{n2}_{r1} exp\left\{ \frac{\Delta G^{00}_M}{RT} \right\}$ <br> $\Delta G^{00}_M = n_1 \mu^*_{a1} + n_2 \mu^*_{r1} - \mu^{00}_M$ |
| **Reverse Micellization Author** | **Equation** | | |
| (25) Pan and Firoozabadi | $\Delta G^{00}_m =$ <br> $\sum\limits_{i=1}^{N_a} \left[ \left(\Delta G^0_a\right)_{Tr} + \left(\Delta G^0_a\right)_{Def} \right] + \left(\Delta G^0_m\right)_{Inf} + \left(\Delta G^0_r\right)_{Tr} + \left(\Delta G^0_r\right)_{Adp} + \left(\Delta G^0_r\right)_{Def}$ | | |

### 2.2.1. Chemical Potential

Chemical equilibrium between two phases is related to the chemical potential and is equal in both phases for each component. Since resins act in the peptizing of asphaltenes, its chemical potential is equal to the resins in the asphaltene and oil phases. The next model was developed according to this thermodynamic concept.

(23) Leontaritis and Mansoori model [39]. The model is based on the thermodynamic-colloidal approach and is capable of predicting the onset of flocculation of colloidal asphaltenes in oil mixtures, either due to changes in composition (solvent addition) or electrical phenomena (streaming potential generation due to flow of asphaltenes containing

oil in conduits or porous media). The model can make predictions regarding the velocity ranges where colloidal asphaltene flocculation can be avoided.

### 2.2.2. Micellization

Micellization is a self-association phenomenon that occurs on the surface of active materials in aqueous systems. An asphaltene colloidal particle has a core of asphaltene molecules surrounded by resin molecules on its surface. The following model is based on this approach.

(24) Victorov and Firoozabadi model [40]. The model describes asphaltene molecules in crude oils as micelles, and the solubilization of asphaltene polar species by resin polar molecules in the micelles. The model describes the change in precipitation power of different alkane precipitants and the effect of pressure on asphaltene precipitation. The amount and the onset of predicted asphaltene precipitation are sensitive to the quantity of resins in the crude oil. The authors concluded that, at high solvent ratios, the asphaltene material does not precipitate, and when precipitation does take place, most of the asphaltene material remains in the crude oil.

### 2.2.3. Reverse Micellization

Reverse micellization assumes that asphaltenes can be redissolved into the oil phase when conditions are favorable for redissolution, and the asphaltene precipitation process is reversible. The following model is based on this approach.

(25) Pan and Firoozabadi model [41,42]. The model was developed using a liquid-liquid equilibrium to obtain a thermodynamic micellization model. The heavy phase is assumed to be in the liquid state and to consist of only asphaltene and resin. The model is used to calculate asphaltene precipitation and shows good accuracy. The effect of pressure, temperature, and composition on precipitation is calculated by the model. The model can calculate resin precipitation at high propane concentrations and asphaltene precipitation at high concentrations of $CO_2$ and injected gasses. The model shows that an increase in resin concentration could inhibit asphaltene precipitation.

## 3. Results and Discussion

Following a consecutive description of the various models mentioned in the preceding section, we use the numbers 1 to 25 to refer to model features and results. The results of statistical analysis refer to the literature experimental data of precipitated asphaltene mass fraction of the reported models.

### 3.1. Experimental Titrations

Liquid titrations are used to determine the amount of asphaltenes for a particular n-alkane titrant/test crude oil system in the laboratory. Calculation of the entire titration curve (i.e., asphaltene weight % vs. volume of n-alkane titrant added/g of test crude oil) is an important calculation test for any asphaltene model. Table 5 shows the type of test crude oils, solvents used, and the type of titration data (liquid/gas) used in the works where a particular model has been reported in the literature. Some models were developed from a set of self-measured experiments, and others have used data reported elsewhere in the literature. As can be seen, the set of experimental data measured by Hirschberg et al. and Burke et al. were used in the development of models 13, 19, 20, 23, 24, and 25.

**Table 5.** Samples, solvents, and methods for experimental titrations for asphaltene precipitation.

| Model | Samples | Solvent | Method | Range of Asphaltene Content (wt%) |
|---|---|---|---|---|
| 1 | Two crude oils (30.54–34.97 °API) | $C_1$, $C_3$, $n-C_5$, $n-C_7$, $n-C_{10}$, $CO_2$ | IP 143 | 0.60 to 3.90 |
| 2 | Six crude oils (19.0–48.0 °API) | $n-C_7$ | ASTM D893-80 with $n-C_7$ | 0.40 to 16.80 |
| 3 | Crude oil (29.0 °API) | $n-C_6$, $n-C_{10}$, $CO_2$ | $n-C_6$, $n-C_{10}$, $CO_2$ precipitations | 5.50 |
| 4 | Light tank crude oil (29.7 °API) | $n-C_5$ to $n-C_{10}$ | IP 143 and Thin Layer Chromatography | 2.20 |
| 5 | Crude oils (19.0–41.0 °API) | $n-C_7$ | $n-C_7$ precipitation and Refractive Index measurements | 1.20 to 10.90 |
| 6 | From the bottom-hole of a production well in San Andres Unit of Seminole oilfield (TX) | $n-C_5$ | $n-C_5$ precipitation | – |
| 7 | Villafortuna-Trecate oil (41.2 °API) and crude oil (35.6 °API) | $n-C_5$ | IP 143 | 0.10 to 1.40 |
| 8 | Light crude oils (40° API) | $n-C_7$ | $n-C_7$ precipitation | 0.30 |
| 9 | Athabasca Bitumen, Cold Lake, Lloydminster | $n-C_5$, $n-C_7$, $n-C_{10}$ | ASTM D2007M | 14.60 to 15.30 |
| 10 | Syncrude coker feed Athabasca bitumen | $n-C_7$ | $n-C_7$ precipitation | 14.50 |
| 11 | Keg river crude oil, Nisku crude oil | $C_2$, $C_3$, $n-C_4$ | $C_2$, $C_3$, and $n-C_4$ precipitations | – |
| 12 | Mars-Pink crude oil | $n-C_5$ to $n-C_{15}$ | $n-C_5$, $n-C_{15}$ precipitations and Refractive Index measurements | 4.40 |
| 13 | Crude oils (19.0–48.0 °API) | $C_1$, $C_3$, $n-C_5$, $n-C_7$, $n-C_{10}$, $CO_2$ | IP 143, ASTM D893-80 with $n-C_7$ | 1.90 to 7.80 |
| 14 | Athabasca Bitumen, Cold Lake, Lloydminster, Venezuelan 1 and 2, Russian, Indonesian | $n-C_5$, $n-C_6$, $n-C_7$, $n-C_8$ $n-C_{10}$ | ASTM D2007M | 4.70 to 21.80 |
| 18 | Crude oils (24.6–44.2 °API) | $n-C_7$, $CO_2$ | $n-C_7$, $CO_2$ precipitations | 0.40 to 4.90 |
| 19 | Crude oil (19 °API), Iranian oil field, crude oil (29 °API) | $CO_2$ | $CO_2$ precipitation | |
| 20 | Crude oils (30.54 °API) | $n-C_5$, $n-C_7$, and $n-C_{10}$ | IP 143 | 4.02 |
| 23 | Two crude oils (30.54–34.97 °API) | $C_1$, $C_3$, $n-C_5$, $n-C_7$, $n-C_{10}$, $CO_2$ | IP 143 | 0.60 to 3.90 |
| 24 | Crude oils (19.0–48.0 °API) | $C_1$, $C_3$, $n-C_5$, $n-C_7$, $n-C_{10}$, $CO_2$ | IP 143, ASTM D893-80 with $n-C_7$ | 0.40 to 16.80 |
| 25 | Crude oil (34.97 °API), North Sea reservoir, Weyburn reservoir | $C_3$, $CO_2$ | IP 143 | 0.60 to 8.90 |

From Table 5, it can be seen that most of the models were developed with crude oils with API gravity ranging from 19 to 48, and only a few used Athabasca bitumen samples. The asphaltene content ranged from 0.1 to 21.8 wt%. Interestingly, the method of choice used to precipitate asphaltenes was IP-143. Some models were developed with precipitated asphaltenes with only one solvent (n-$C_5$, n-$C_7$ or $CO_2$), while others used up to six different solvents.

### 3.2. Pressure and Temperature Conditions for Each Model

Since each model was developed with different types of crude oil and solvent, the pressure and temperature conditions for asphaltene precipitation were different, and the applicability of each model depends on all of these conditions. Figure 1 depicts the pressure and temperature conditions used to develop each model. Not all of the models reported the experimental conditions. In general, temperatures ranged from $-20$ to $250\ ^\circ$C, while pressure ranged from 0 to 105.5 MPa. Only model 11 used a wide range of temperatures, i.e., $-20\ ^\circ$C to $250\ ^\circ$C, being the one with the widest temperature range. Models 7 and 19 were second in terms of wider temperature ranges (25–177 $^\circ$C), and the widest pressure range (0–105 MPa for model 7, and 5–50 MPa for model 19). The model with the narrowest pressure and temperature ranges was model 4 (21 to 38 $^\circ$C, and 0.1 to 1.0 MPa).

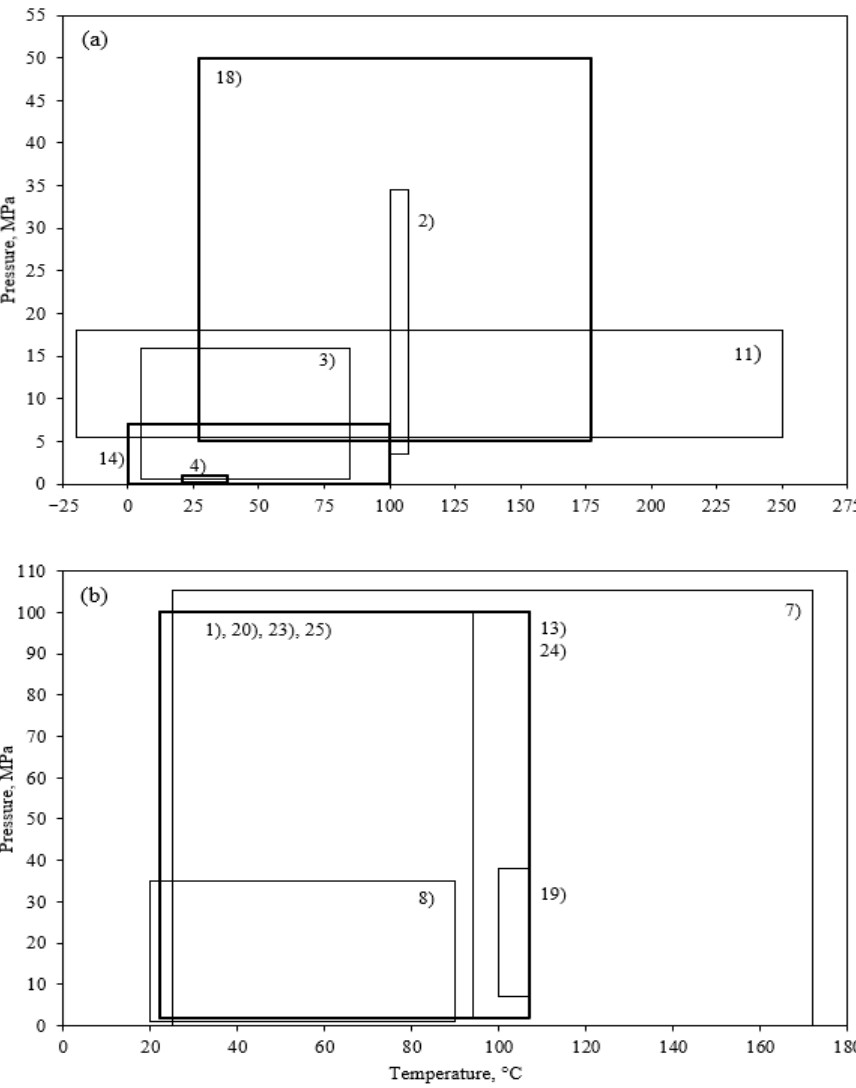

**Figure 1.** Pressure and temperature conditions for different models. (**a**) models: 2, 3, 4, 11, 14, and 18. (**b**) models: 1, 7, 8, 13, 19, 20, 23, 24, and 25.

### 3.3. Models Excluded from the Analysis

Some models did not report information to validate their accuracy in calculating asphaltene precipitation (models 5, 8, 10, 12, 21, and 23), and others only reported limited information (models 2, 3, 6, 7, 9, 11, 13, and 22). For later models, the limited information provided was considered insufficient to carry out a statistical analysis. Only 11 out of the 25 models (i.e., 1, 4, 13, 14, 16, 17, 18, 19, 20, 24, and 25) reported information on the precipitated asphaltene mass fraction to perform the statistical analysis. Models excluded from the analysis were: 2, 3, 5, 6, 7, 8, 9, 10, 11, 12, 15, 21, 22, and 23. The few calculation properties reported by the models excluded were: solubility parameters for asphaltenes and solvents (Burke et al., Chung et al., and de Boer et al. models), refractive index of asphaltenes and solvents (Buckley et al., Ting et al., and Wang and Buckley models). Furthermore, a comparison between experimental and calculated asphaltene precipitation data was not reported by these models.

### 3.4. Analysis of Models

Asphaltenes are dispersed in crude oil in equilibrium with saturates, aromatics, and resins fractions, constituting a colloidal system. Asphaltene precipitation is caused due to changes in pressure, temperature, composition, flow, etc., which alter this colloidal system equilibrium, and then induce aggregation and precipitation.

#### 3.4.1. Effects of Pressure and Temperature

Hirschberg et al. and Pan and Firoozabadi reported comparisons of calculated values from their models against experimental data from Hirschberg et al., (tank oil 1 and propane weight ratio 1:7, at 93 °C) as depicted in Figure 2a. In general, models calculated the trend of asphaltenes precipitation with changes in pressure well; i.e., when pressure decreases, asphaltene precipitation increases. Shirani et al., Ngheim et al., and Victorov and Firoozabadi reported comparisons of calculated values from their models against experimental data from Burke et al., (live crude oil 3 at 100 °C) as depicted in Figure 2b. The model by Shirani et al. could calculate asphaltene precipitation with higher accuracy than the models by Nghiem et al. and Victorov and Firoozabadi. It exhibited a slight error at pressures between 13 and 15 MPa.

At 30 °C, Figure 3 shows that Li and Firoozabadi´s model was not able to calculate asphaltene precipitation from the experimental data from Szewczyk et al. [43,44]. At low pressures (<10 MPa), the model then overestimated the experimental data from 10 to 15 MPa for oil X2. For pressures higher than 25 MPa, it calculated the experimental data quite well, with a slight underestimation. For oil X3, the model was not able to calculate asphaltene precipitation at P <20 MPa, and it overestimated the experimental data from pressures 20 to 25 MPa. Above 30 MPa, the model showed a slight underestimation.

#### 3.4.2. Effects of Injected $CO_2$

Injection of $CO_2$ is one of the techniques used for oil recovery, but sometimes it can lead to significant problems of asphaltene precipitation in the reservoir. Figure 4 shows that Li and Firoozabadi, Pan and Firoozabadi, Wu et al., and Shirani et al. models calculated that, as $CO_2$ concentration increases, asphaltene precipitation also increases. Experimental data from Weyburn crude oil with 4.9 wt% of asphaltenes [45] were used by these models to obtain asphaltene precipitation in the presence of $CO_2$. The models by Pan and Firoozabadi and Li and Firoozabadi exhibited a tendency to underestimate the experimental data from 0.41 to 0.6 fractions of $CO_2$ and overestimated the experimental data up to 0.65 to 0.75 fractions of $CO_2$ with a maximum error of 29%. Pan and Firoozabadi were not able to calculate asphaltene precipitation at <0.51 mole fraction of $CO_2$. It exhibited a maximum error of almost 15%. The model by Shirani et al. exhibited good accuracy in reproducing experimental data quite well from 0.46 to 0.53 mole fractions of $CO_2$, then underestimated the experimental data from 0.54 to 0.65 mole fractions of $CO_2$. It exhibited a maximum error of 17%. The model by Wu et al. overestimated the experimental data

up to 0.38 to 0.54 fractions of $CO_2$ and underestimated the experimental data from 0.55 to 0.65 with a maximum error of 23%.

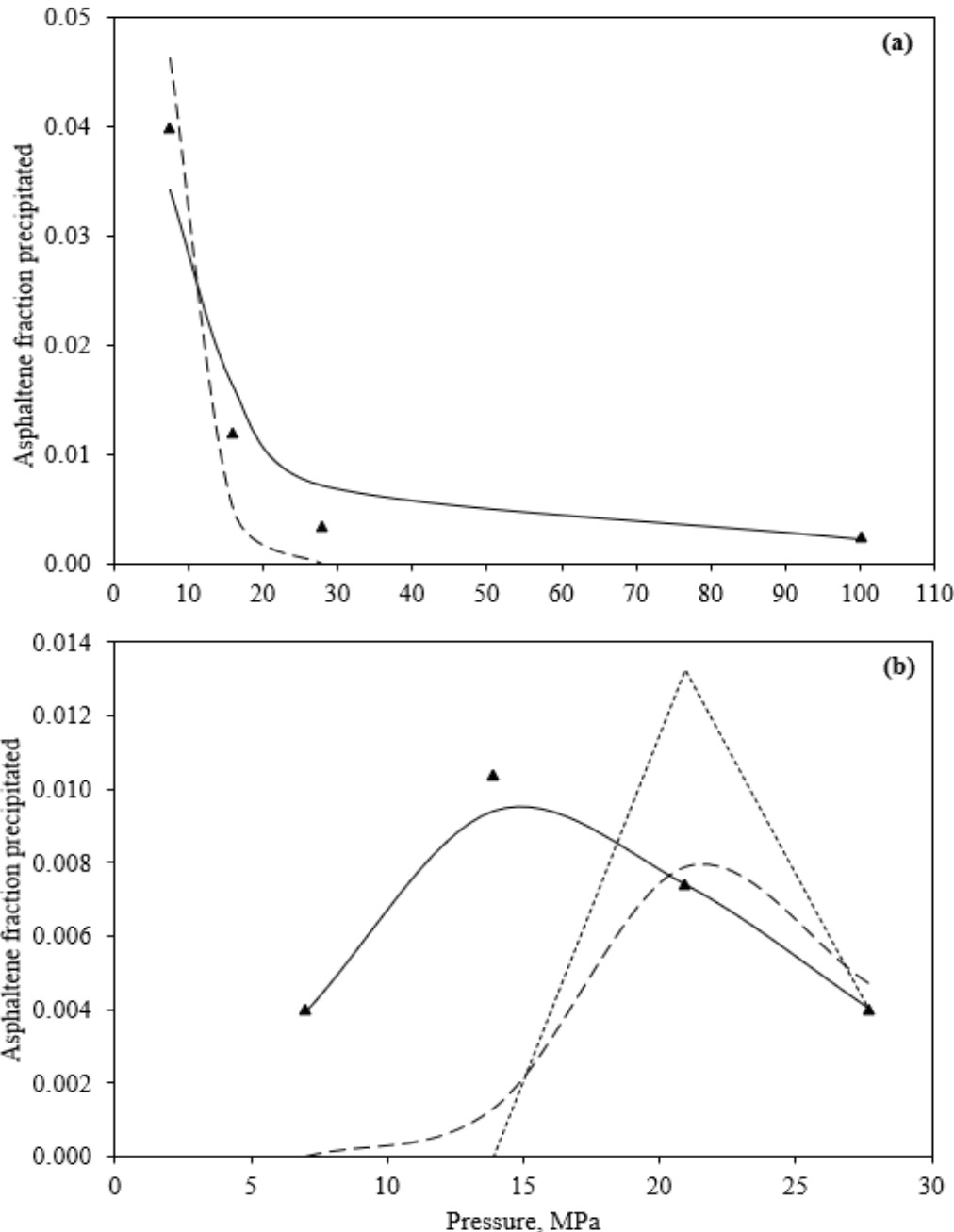

**Figure 2.** Pressure and temperature effects on precipitated asphaltene fraction calculated by different models. (**a**) Experimental data (▲) by Hirschberg et al., at 93 °C, (—) calculated by Pan and Firoozabadi) and (— —) calculated by Hirschberg et al.). (**b**) Experimental data (▲) by Burke et al., at 100 °C, (—) calculated by Shirani, (— — —) calculated by Victorov and Firoozabadi) and (— —) calculated by Nghiem.

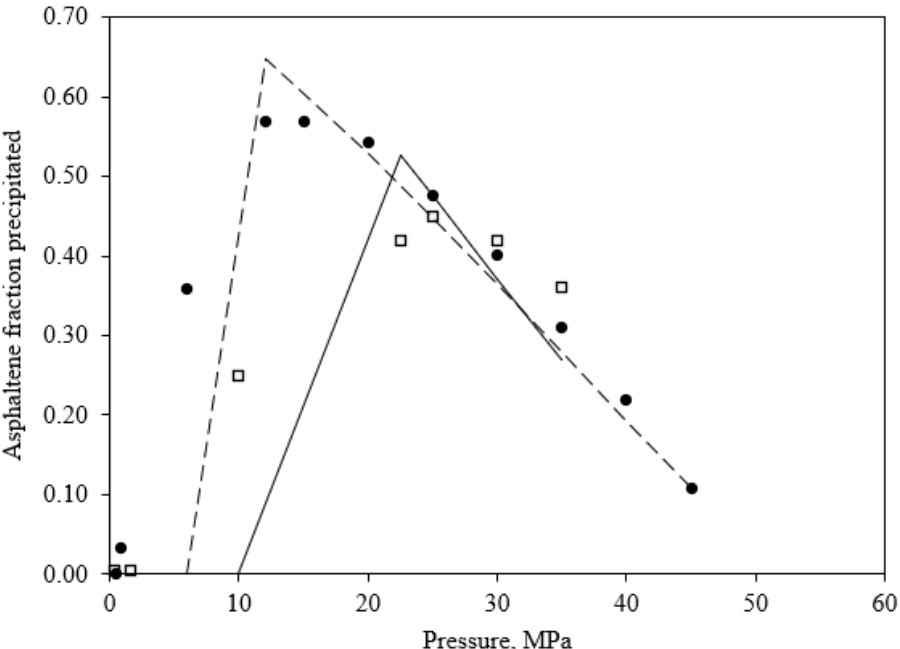

**Figure 3.** Pressure and temperature effects on precipitated asphaltene fraction calculated by Li and Firoozabadi. Experimental data (●) by Szewczyk et al., at 30 °C for oil X2, (— —) calculated. Experimental data (□) by Szewczyk et al., at 30 °C for oil X3 (—) calculated.

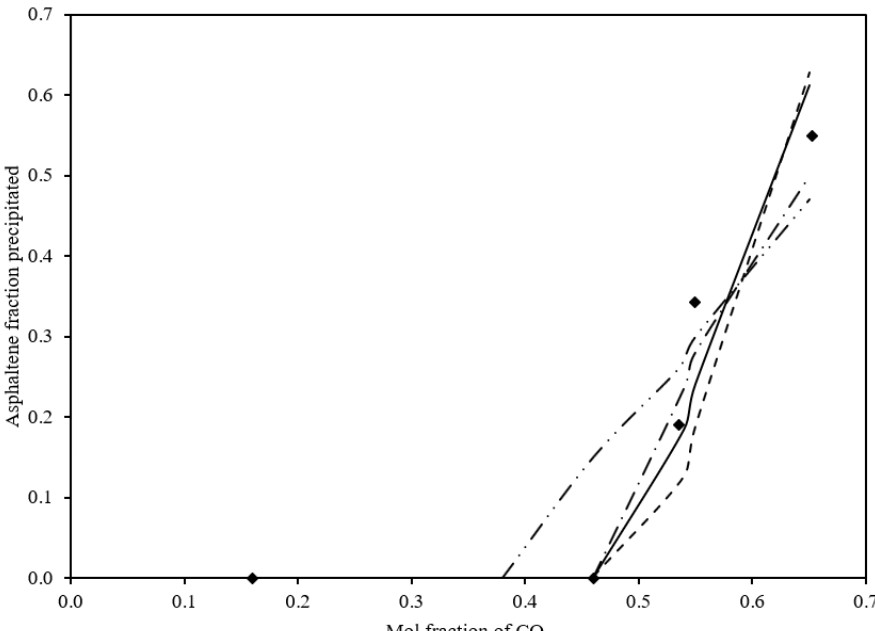

**Figure 4.** Effect of $CO_2$ injection (in Weyburn oil) on precipitated asphaltene fraction. Experimental data (◆), (— ·· —) calculated by Wu et al., (— · —) calculated by Shirani et al., (—) calculated by Pan and Firoozabadi, and (— —) calculated by Li and Firoozabadi.

### 3.4.3. Effects of the Addition of n-Alkanes as Solvents

The models by Kawanaka et al., Sabbagh et al., Rassamdana et al., Wu et al., Buenrostro et al., and Victorov and Firoozabadi reported asphaltene precipitation by adding n-C5 and n-C7 as solvents. Figures 5 and 6 depict the experimental data versus the calculated results for these models. The models show an increase in asphaltenes precipitation due to an increase in solvents (n-C5 and n-C7). Figure 5a,b shows the results of asphaltene precipitation in five models with n-C5 as solvent. The model by Kawanaka et al. used

the experimental data of crude oil with a 1.9 wt% asphaltene content. Initially, the model overestimated the experimental data from 0.60 to 0.80 mass fraction of the solvent, then calculations exhibited good accuracy in the range of 0.85 to 0.97 mass fraction of the solvent. It exhibited a maximum error of 3%. Sabbagh et al. calculated asphaltene precipitation from Athabasca bitumen and Cold Lake bitumen with contents of 14.60 and 15.30 wt% of asphaltenes, respectively. For both samples, the model underestimated the experimental data, and presented a maximum error of 51% for Athabasca bitumen and 27% for Cold Lake bitumen. Rassamdana et al. and Victorov and Firoozabadi could calculate experimental data with a maximum error of 42 and 29%, respectively. Wu et al. could calculate asphaltene precipitation quite well for Suffield oil. For Lindberg oil, the model exhibited a considerable deviation from 0.6 to 0.94 mass fraction of n-C5. In general, the model showed a maximum error of 14%. Buenrostro et al. were able to reproduce experimental data quite well for both Mexican oils (C1 and Y3). The model exhibited a slight underestimation at 0.97 mass fraction of n-C5.

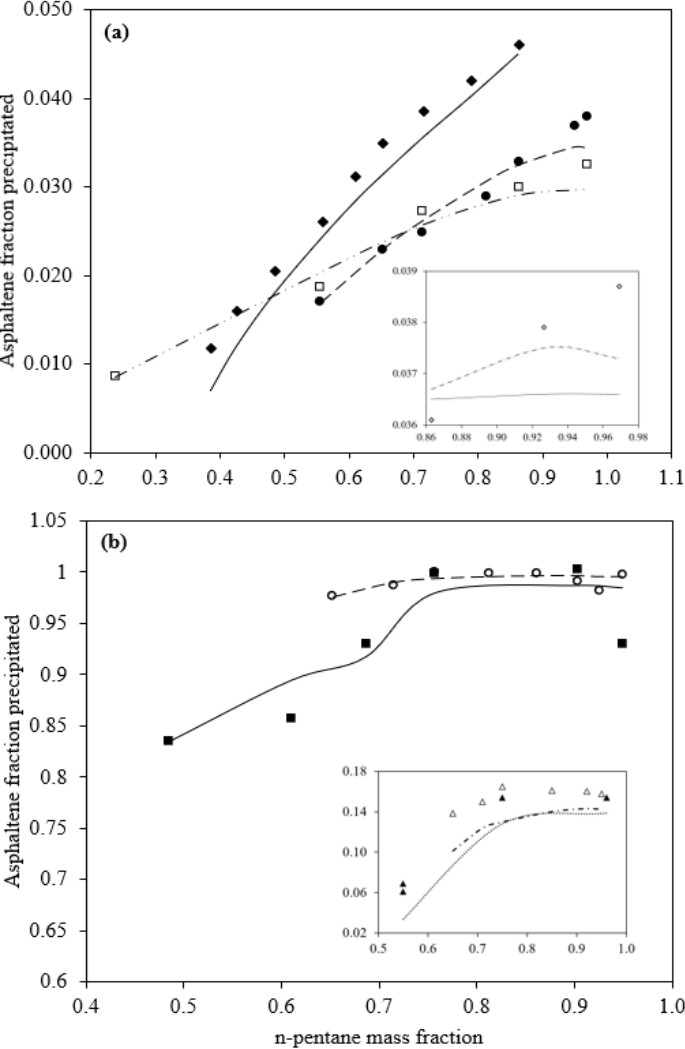

**Figure 5.** Effect of adding n-pentane solvent on precipitated asphaltene fraction. (**a**) Rassamdana et al., (◆ experimental, —— calculated), Buenrostro et al., (for oil Y3: □ experimental, — ·· — calculated, and for oil C1: ● experimental, —— calculated), plot inside refers to: Experimental data (▲) by Hirschberg et al., at 60 °C, (●●●●) calculated by Victorov and Firoozabadi, and (— · —) calculated by Kawanaka et al., (**b**) Wu et al., (for oil Lindberg: ■ experimental, — calculated, and for oil Suffield: ○ experimental, — — calculated), plot inside refers to Sabbagh et al., (for Athabasca bitumen: ▲ experimental, ●●●● calculated, and for Cold lake bitumen: △ experimental, — · — calculated).

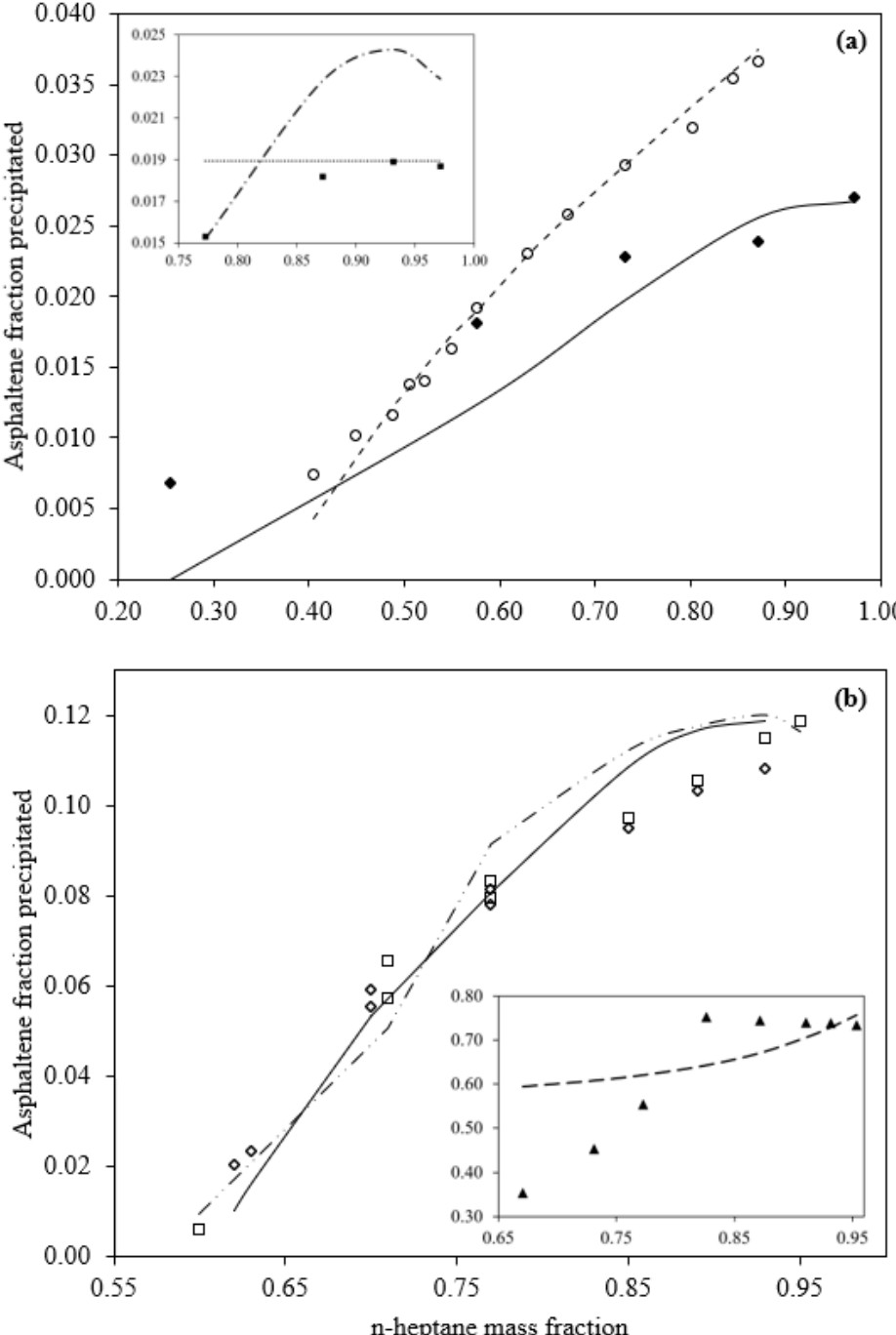

**Figure 6.** Effect of adding n-heptane solvent on precipitated asphaltene fraction. (**a**) Buenrostro et al., (for oil Y3: ◆ experimental, —— calculated), Rassamdana et al., (○ experimental, — — calculated), plot inside refers to: Experimental data (■) by Hirschberg et al., at 60 °C, (••••) calculated by Victorov and Firoozabadi, and (— · —) calculated by Kawanaka et al., (**b**) Sabbagh et al., (for Athabasca bitumen: □ experimental, —··— calculated, and for Cold lake bitumen: ◇ experimental, —— calculated), plot inside refers to: Wu et al., (for Suffield oil: ▲ experimental, — — calculated).

Figure 6a,b shows the results of asphaltene precipitation by five models with n-$C_7$ as a solvent. The model by Kawanaka et al. calculated the experimental data quite well from 0.60 to 0.80 mass fraction of n-$C_7$, then overestimated the experimental data above 0.80 mass fraction of n-$C_7$. The model by Sabbagh et al. initially exhibited good accuracy from 0.60 to 0.71 mass fraction of n-$C_7$, then, for both samples, the model overestimated the experimental data from 0.71 to 0.95 mass fraction of n-$C_7$. The model exhibited a maxi-

mum error of 51% for Athabasca bitumen, and 50% for Cold Lake bitumen. The model by Rassamdana et al. calculated the experimental data with a certain accuracy in all ranges of mass fraction of n-$C_7$, and exhibited a maximum error of 42%. The model by Victorov and Firoozabadi presented a constant amount of precipitated asphaltene (0.019 wt% of asphaltene) from 0.78 to 0.98 mass fraction of n-$C_7$. The model by Wu et al. was capable of calculating asphaltene precipitation quite well for Suffield oil. In general, the model showed a maximum error of 14%. The model by Buenrostro et al. underestimated experimental data from 0.25 to 073 mass fraction of n-$C_7$ and exhibited a slight overestimation at 0.87 mass fraction of n-$C_7$.

*3.5. Statistical Analysis*

The experimental data sets on the precipitated asphaltene mass fraction reported by some models were used to perform a statistical analysis to determine which model is the most accurate to calculate asphaltene precipitation according to the effect of pressure, $CO_2$ injection, and addition of n-alkanes (n-$C_5$ and n-$C_7$). To evaluate the accuracy of each model, the following parameters were used:

$$\text{Residual } (R): \ R = x_{wa}^{exp} - x_{wa}^{calc} \tag{1}$$

$$\text{Error } (E): \ E = \left( \frac{x_{wa}^{exp} - x_{wa}^{calc}}{x_{wa}^{exp}} \right) \times 100 \tag{2}$$

$$\text{Average absolute relative deviation } (AARD): \ AARD_i = \frac{100}{n} \sum \left| \frac{x_{wa}^{exp} - x_{wa}^{calc}}{x_{wa}^{exp}} \right| \tag{3}$$

Tables 6–8 show the results of the statistical analysis of the models.

**Table 6.** Statistical analysis for models to calculate asphaltene precipitation by effects of pressure and temperature.

| Model | Hirschberg et al. 1 | Nghiemet al. 13 | Li and Firoozabadi 18 | Shirani et al. 19 | Victorov and Firoozabadi 24 | Pan and Firoozabadi 25 |
|---|---|---|---|---|---|---|
| AARD, % | 74.8564 | 52.6946 | 40.3938 | 7.3951 | 70.0516 | 40.8937 |
| (+) Residuals | 4 | 2 | 13 | 8 | 3 | 3 |
| (−) Residuals | 1 | 2 | 4 | 7 | 1 | 5 |
| Highest positive residual | 0.0070 | 0.0091 | 0.3588 | 0.0093 | 0.0104 | 0.0057 |
| Lowest negative residual | −0.0064 | −0.0007 | −0.1058 | −0.0034 | −0.0058 | −0.0044 |
| Range | 0.0134 | 0.0097 | 0.4646 | 0.0127 | 0.0162 | 0.0101 |
| $R^2$ | 0.9817 | 0.0014 | 0.7902 | 0.9857 | 0.0028 | 0.9532 |
| Slope | 1.2743 | 0.0434 | 1.0470 | 0.9714 | 0.1092 | 0.8243 |
| Intercept | −0.0057 | 0.0032 | −0.0522 | 0.0000 | 0.0036 | 0.0025 |

Statistical Analysis for the Best Model to Calculate Asphaltene Precipitation

(a) For pressure and temperature effects. The models by Hirschberg et al., Nghiem et al., Li and Firoozabadi, Shirani et al., Victorov and Firoozabadi, and Pan and Firoozabadi were analyzed. The statistical analysis for the models evaluated are presented in Table 6. The model by Shirani et al. exhibited the best AARD% value (7.39%) and a good balance of residuals with eight positive residuals and seven negative residuals, and values of R2, slope, and intercept of 0.985, 0.9714, and 0.00, respectively. The model by Hirschberg et al. showed a tendency to underestimate the experimental data, as it presented four positive residuals and only one negative residual, and an AARD% value of 74.85%. According to the parameters of R2, slope, and intercept, Victorov and Firoozabadi´s model exhibited

values of 0.0028, 0.1092, and 0.0036, being the worst model to calculate asphaltene precipitation due to changes in pressure and temperature.

**Table 7.** Statistical analysis for models to calculate asphaltene precipitation by effects of $CO_2$ injection.

| Model | Wu et al. 16 | Li and Firoozabadi 18 | Shirani et al. 19 | Pan and Firoozabadi 25 |
|---|---|---|---|---|
| AARD, % | 23.5848 | 29.4782 | 17.2113 | 14.5783 |
| (+) Residuals | 2 | 2 | 2 | 2 |
| (−) Residuals | 1 | 1 | 1 | 1 |
| Highest positive residual | 0.0800 | 0.1492 | 0.0622 | 0.1017 |
| Lowest negative residual | −0.0800 | −0.0741 | −0.0463 | −0.0624 |
| Range | 0.1600 | 0.2233 | 0.1085 | 0.1642 |
| $R^2$ | 0.9312 | 0.9036 | 0.9237 | 0.9030 |
| Slope | 0.5799 | 1.3970 | 0.7508 | 1.2222 |
| Intercept | 0.1365 | −0.1863 | 0.0679 | −0.0949 |

**Table 8.** Statistical analysis for models to calculate asphaltene precipitation by effects of adding n-alkanes (n-$C_5$ and n-$C_7$), as precipitants.

| Model | Rassamdana et al. 4 | Sabbagh et al. 14 | Wu et al. 16 | Buenrostro et al. 17 | Kawanaka et al. 20 | Victorov and Firoozabadi 24 |
|---|---|---|---|---|---|---|
| AARD, % | 9.5662 | 18.0057 | 7.5443 | 13.0570 | 13.8824 | 15.4236 |
| (+) Residuals | 15 | 18 | 11 | 11 | 3 | 6 |
| (−) Residuals | 7 | 10 | 9 | 5 | 4 | 1 |
| Highest positive residual | 0.0048 | 0.0371 | 0.1099 | 0.0069 | 0.0014 | 0.0056 |
| Lowest negative residual | −0.0025 | −0.0151 | −0.2413 | −0.0018 | −0.0054 | −0.0005 |
| Range | 0.0073 | 0.0522 | 0.3512 | 0.0086 | 0.0068 | 0.0061 |
| $R^2$ | 0.9728 | 0.8745 | 0.8743 | 0.9307 | 0.3049 | 0.9815 |
| Slope | 1.0417 | 0.8781 | 0.7686 | 1.0427 | 0.3458 | 1.1614 |
| Intercept | −0.0020 | 0.0043 | 0.2098 | −0.0023 | 0.0208 | −0.0072 |

(b) For $CO_2$ injection effect. The models by Wu et al., Li and Firoozabadi, Shirani et al., and Pan and Firoozabadi were analyzed. Table 7 exhibits statistical analyses realized for these models. The model by Pan and Firoozabadi et al. exhibited the best AARD% value, with 14.578%, and a good balance of residuals with two positive residuals and one negative residuals. This means that the model does not tend to over or underestimate the experimental data. The model that exhibited the highest value of AARD% was the model by Li and Firoozabadi, with a value of 29.478%. According to the parameters of R2, slope, and intercept, the model by Wu et al. exhibited the best values (0.931, 0.579, and 0.136, respectively). The model by Shirani et al. exhibited the best performance to calculate asphaltene precipitation with pressure, as well as temperature changes. The model by Pan and Firoozabadi exhibited the best performance to calculate asphaltene precipitation with an injection of $CO_2$ effects.

(c) For n-$C_5$ and n-$C_7$ titration tests. The models by Rassamdana et al., Sabbagh et al., Wu et al., Buenrostro et al., Kawanaka et al., and Victorov and Firoozabadi were analyzed. According to residual balance, the model by Rassamdana et al. did not over or underestimate the experimental data. It exhibited a residual balance of fifteen positive residuals and seven negative residuals. The model by Sabbagh et al. exhibited 18 positive residuals and 10 negative residuals, indicating that the model did not tend to over or underestimate the experimental data when n-$C_5$ or n-$C_7$ were added. The few experimental data reported by Kawanaka et al. and Victorov and Firoozabadi were not sufficient (seven experimental data were reported for both models) to determine their accuracy. The model

by Kawanaka et al. exhibited a residual balance of three and four positive and negative residuals, while the model by Victorov and Firoozabadi showed six and one positive and negative residuals, respectively. The models by Wu et al. and Buenrostro et al. exhibited the best residual balance, with eleven positive residuals and nine negative residuals, and eleven positive residuals and five negative residuals, respectively. According to AARD%, the model by Wu et al. was the best, with 7.54%. The model by Rassamdana et al. was second with an AARD% value of 9.56%, and the model by Victorov and Firoozabadi exhibited an AARD% value of 15.42%, the highest value in this analysis. For parameters $R^2$, slope, and intercept, the models by Rassamdana et al. and Buenrostro et al. exhibited similar values (0.972, 1.041 and $-0.002$ for Rassamdana, 0.930, 1.042 and $-0.002$ for Buenrostro), being the models with best performance to calculate asphaltene precipitation due to the addition of n-alkanes (n-$C_5$ and n-$C_7$).

In order to determine which model was the best in calculating asphaltene precipitation among all models analyzed, we used the average absolute relative error (AARD, %). Tables 6–8 and Figure 7 depict values of AARD for each model analyzed. The model by Shirani et al. exhibited the lowest values, with 7.395% for pressure and temperature changes. The model by Pan and Firoozabadi exhibited the lowest value, with 14.578% for the injection of $CO_2$. The model by Wu et al. exhibited the lowest value, with 7.544% for the addition of n-$C_5$/n-$C_7$. The model by Hirschberg et al. exhibited the highest value (74.856%) for pressure and temperature changes. The model by Li and Firoozabadi exhibited the highest values (29.478%) for $CO_2$ injection effect. The model by Sabbagh et al. exhibited the highest values (18.005%) for the addition of n-$C_5$/n-$C_7$ effect. For $R^2$, slope, and intercept, the model by Shirani et al. exhibited the best values (0.985, 0.971, and 0.0003, respectively) for pressure and temperature changes. The model by Wu et al. exhibited the best values (0.931, 0.579, and 0.136, respectively) for the injection of $CO_2$ effect. The model by Victorov and Firoozabadi exhibited the best values (0.981, 1.161, and $-0.007$, respectively) for addition of n-$C_5$/n-$C_7$ effect.

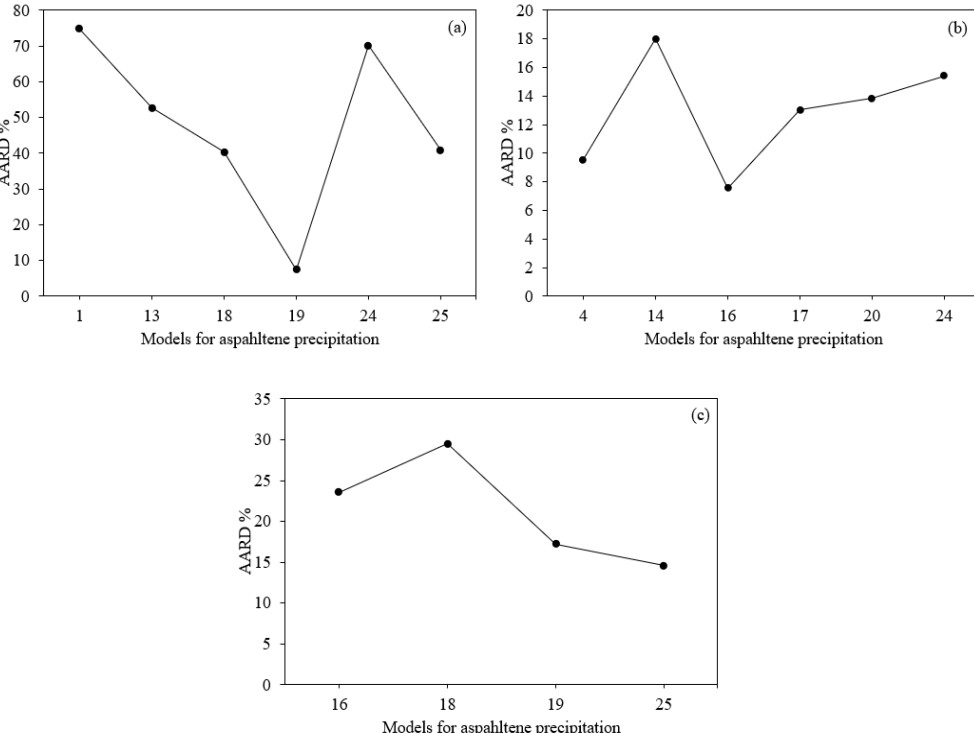

**Figure 7.** AARD% for different models to calculate asphaltene precipitation. (**a**) by effect of pressure and temperature changes. (**b**) by effect of addition of n-$C_5$/n$C_7$. (**c**) by effect of $CO_2$ injection.

## 4. Discussion

In this work, a study of the calculation capabilities of eleven published models of asphaltene precipitation was carried out, using statistical analysis from their reported calculation results in the literature. Model accuracies were compiled and used to perform a statistical analysis to determine which model could calculate asphaltene precipitation with good accuracy. It was observed that when the pressure is reduced in a system, the precipitation of asphaltenes is favored; the temperature also increases the amount of asphaltenes to precipitate decreases.

In the case of $CO_2$ and n-$C_5$ and n-$C_7$ titrations, the precipitation of asphaltenes increases when the addition of these compounds increases. This is due to the low or no solubility of asphaltenes with saturated compounds (n-alkanes). Unlike other models, the model by Sabbagh et al. explicitly addresses heavy crude oils (Athabasca and Cold Lake bitumen). This model only covers the addition of n-$C_5$ and n-$C_7$ as solvents.

Table 5 shows the types of crude oils used for the development of each model, as well as the solvent used in its development and the amount of asphaltenes present in the sample. On the other hand, Figure 1 shows the pressure and temperature conditions for each model.

As seen in Table 5, previous studies have focused on experiments with blends of crude oils with aromatic solvents (i.e., toluene) and precipitants (i.e., n-$C_5$, n-$C_7$, and $CO_2$). Blending crude oils with different chemical natures may cause asphaltene precipitation, so there is still a need to improve some of the models so that their applicability is extended to heavier crude oils, as well as for mixtures between different types of crude oils.

To achieve these goals, it is important to obtain accurate and exhaustive experimental data so that the parameters included in the models can be properly predicted to allow for better predictions of the behavior of hydrocarbons under different conditions. Experiments with different crude oils and asphaltenes are also necessary to establish a dependence of model parameters on feed properties.

## 5. Conclusions

From the literature reports, it is observed that the solubility approach has a greater impact on the study of asphaltene precipitation, with 18 models developed under this method. Most of the models in the literature have used n-$C_5$ and n-$C_7$ titration data to determine the model's performance and accuracy. Since the solubility and colloidal approaches require the calculation of thermodynamic properties, such as solubility parameters, molar volumes and molecular weight, equations-of-state, such as Soave-Redlich-Kwong and Peng-Robinson (Hirschberg et al., Novosad and Costain, Rassamdana et al., Cimino et al., Nghiem et al., Sabbagh et al., Li and Firoozabadi, Shirani et al., Ting et al., Wu et al., Buenrostro et al.), are typically used to determine these thermodynamic parameters for crude oils and solvents. According to the statistical analysis of the precipitated asphaltene mass fraction by different procedures and approaches, the model by Shirani et al. exhibited the best values for $R^2$, slope, and intercept (0.985, 0.971, and 0.0003, respectively), as well as an AARD value of 7.395% when this model was applied to calculate the amount of precipitated asphaltenes as functions of pressure and temperature changes. The model by Victorov and Firoozabadi exhibited values for $R_2$, slope, and intercept of 0.981, 1.161, and −0.007 when this model was applied to calculate the amount of precipitated asphaltenes due to the addition of n-alkanes. The model by Wu et al. exhibited values for $R^2$, slope, and intercept of 0.931, 0.5799, and 0.136 when this model was applied to calculate the amount of precipitated asphaltenes due to the injection of $CO_2$. As per the AARD analysis, models that exhibited the worst values of AARD were those by Hirschberg et al., with 74.85%, and Victorov and Firoozabadi, with 70.05% for pressure and temperature changes. The models by Li and Firoozabadi, with 29.47%, and Wu et al., with 23.58%, had the worst values for $CO_2$ injection effect. The models by Sabbagh et al., with 18.00%, and Victorov and Firoozabadi, with 15.42%, had the worst values for the addition of n-$C_5$/n-$C_7$ effect. From these results, it is evident that it is necessary to develop a model that ad-

dresses the precipitation of asphaltenes when dilutions are made between crude oils with different characteristics.

**Author Contributions:** E.A.H.: Conceptualization; Formal analysis; Investigation; Methodology; Validation; Writing—original draft; Writing—review & editing; C.L.-G.: Investigation; Methodology; Validation; Writing—original draft; J.A.: Methodology; Validation; Writing—original draft; Writing—review & editing. All authors have read and agreed to the published version of the manuscript.

**Funding:** This research received no external funding.

**Data Availability Statement:** Data sharing not applicable. No new data were created or analyzed in this study. Data sharing is not applicable to this article.

**Acknowledgments:** E.A.H. thanks to Consejo Nacional de Ciencia y Tecnología (CONACyT) for the Ph.D. scholarship. The authors also thank the Mexican Institute of Petroleum and CONACyT for the financial support (Project Y.61057).

**Conflicts of Interest:** The authors declare that they have no known competing financial interests or personal relationships that could have appeared to influence the work reported in this paper.

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
