# Peer review of "Analysis of Asphaltene Precipitation Models from Solubility and Thermodynamic-Colloidal Theories"

_processes, doi:10.3390/pr11030765_

Round 1

Reviewer 1 Report

By producing blockage, separation, and other complications, asphaltene are a difficulty in the production and processing of oil. Different models have been used by scientists to try and forecast when these issues may arise, and 24 models have been examined to determine how well they correspond to actual tests. Statistical methods were used to compare the models, and the findings indicated that further work has to be done to make the models more useful under a wider variety of scenarios, such as those involving various pressures, temperatures, and compositions.

Instead of utilizing 24 models, artificial intelligence can solve the problem of selecting a decent model to capture experimental data, although writers might still attempt to draw further conclusions from poor or good model matches with experimental data. 

It is advised to utilise the  follwowing references after to make the introduction more thorough because it lacks depth. 

Alimohammadi, S., Zendehboudi, S., & James, L. (2019). A comprehensive review of asphaltene deposition in petroleum reservoirs: Theory, challenges, and tips. Fuel252, 753-791.

*Moud, A. A. (2022). Asphaltene induced changes in rheological properties: A review. Fuel316, 123372. 

*Adams, J. J. (2014). Asphaltene adsorption, a literature review. Energy & Fuels28(5), 2831-2856.

Reviewer 2 Report

Hernandez et al. showed an exciting analysis of asphaltene precipitation models. It is clear that more work is needed to clearly identify the correct model, EoS, and correlation...to identify asphaltene problems in the well of interest. The work is excellent and I have three minor recommendations.

- Please elaborate on the importance of data for the correct analysis and application of the models. Which is, in your opinion, the critical parameter that should be obtained. For example, in my experience, the onset at real reservoir conditions is a must!

- What do you think about using a generalized EoS for an entire field.

- Please include models that identify precipitation in the reservoir and calculate the deep of asphaltene precipitation in the reservoir. 

Round 2

Reviewer 1 Report

All comments are addressed!